# NK Cell-Based Immunotherapy for Hematological Malignancies

**DOI:** 10.3390/jcm8101702

**Published:** 2019-10-16

**Authors:** Simona Sivori, Raffaella Meazza, Concetta Quintarelli, Simona Carlomagno, Mariella Della Chiesa, Michela Falco, Lorenzo Moretta, Franco Locatelli, Daniela Pende

**Affiliations:** 1Department of Experimental Medicine, University of Genoa, 16132 Genoa, ItalySimona.Carlomagno@unige.it (S.C.); Mariella.DellaChiesa@unige.it (M.D.C.); 2Centre of Excellence for Biomedical Research, University of Genoa, 16132 Genoa, Italy; 3Department of Integrated Oncological Therapies, IRCCS Ospedale Policlinico San Martino, 16132 Genoa, Italy; raffaella.meazza@hsanmartino.it; 4Department of Hematology/Oncology, IRCCS Ospedale Pediatrico Bambino Gesù, 00165 Rome, Italy; concetta.quintarelli@opbg.net (C.Q.); franco.locatelli@opbg.net (F.L.); 5Department of Clinical Medicine and Surgery, University of Naples Federico II, 80131 Naples, Italy; 6Integrated Department of Services and Laboratories, IRCCS Istituto Giannina Gaslini, 16147 Genoa, Italy; michelaemma.falco@gmail.com; 7Department of Immunology, IRCCS Ospedale Pediatrico Bambino Gesù, 00146 Rome, Italy; lorenzo.moretta@opbg.net; 8Department of Gynecology/Obstetrics and Pediatrics, Sapienza University, 00185 Rome, Italy

**Keywords:** NK cells, receptors, acute leukemia, hematopoietic stem cell transplantation, HLA class I, killer immunoglobulin-like receptors, NK cell alloreactivity, cytokines, CAR-NK cells, immunotherapy

## Abstract

Natural killer (NK) lymphocytes are an integral component of the innate immune system and represent important effector cells in cancer immunotherapy, particularly in the control of hematological malignancies. Refined knowledge of NK cellular and molecular biology has fueled the interest in NK cell-based antitumor therapies, and recent efforts have been made to exploit the high potential of these cells in clinical practice. Infusion of high numbers of mature NK cells through the novel graft manipulation based on the selective depletion of T cells and CD19^+^ B cells has resulted into an improved outcome in children with acute leukemia given human leucocyte antigen (HLA)-haploidentical hematopoietic transplantation. Likewise, adoptive transfer of purified third-party NK cells showed promising results in patients with myeloid malignancies. Strategies based on the use of cytokines or monoclonal antibodies able to induce and optimize NK cell activation, persistence, and expansion also represent a novel field of investigation with remarkable perspectives of favorably impacting on outcome of patients with hematological neoplasia. In addition, preliminary results suggest that engineering of mature NK cells through chimeric antigen receptor (CAR) constructs deserve further investigation, with the goal of obtaining an “off-the-shelf” NK cell bank that may serve many different recipients for granting an efficient antileukemia activity.

## 1. Introduction

Natural killer (NK) cells are cytotoxic and cytokine-producing components of innate lymphoid cells (ILCs), playing important roles in antiviral and antitumor defense [1,2]. ILCs represent a heterogeneous group of immune cells that are mainly localized at epithelial surfaces, where they maintain tissue homeostasis and quickly respond to pathogen invasion by mediating appropriate immune responses. They develop from a common lymphoid progenitor but, differently from T and B lymphocytes, lack the expression of antigen receptors encoded by rearranged genes. ILCs can be considered the innate counterparts of T cell subsets. In particular, NK cells represent the “cytotoxic ILC”, whereas ILC1, ILC2, and ILC3 are considered as “helper ILCs” because they are noncytolytic and produce sets of cytokines unique for each subset. Both NK cells and helper ILC1 share the expression of Tbet transcription factor, encoded by the *Tbx21* gene that is involved in IFN-γ production, but differ in eomesodermin (Eomes) transcription factor expression. Indeed, NK cells are Tbet^+^ Eomes^+^ while ILC1 are Tbet^+^ Eomes^−^ [3,4]. Recent advances of our knowledge underline a certain degree of plasticity among the various ILC subsets, mainly by the influence of tissue microenvironment [2,5].

NK cells are equipped with a wide array of germline-encoded inhibitory and activating receptors, which can be engaged by specific ligands expressed on various cells at the immunological synapse. NK cell function is a finely tuned balance between activating and inhibitory signaling transmitted by these receptors. NK cells preserve tolerance towards surrounding healthy cells, mainly through inhibitory receptors recognizing self-major histocompatibility complex (MHC) class I molecules. In humans, they are represented by killer immunoglobulin-like receptors (KIRs) and CD94:natural killer group 2A (NKG2A), specific for classical and nonclassical HLA class I molecules, respectively. In the process of NK cell “education”, the strength of these inhibitory receptor/ligand interactions positively correlates with the functional potential of NK cells [6]. Responsible for the “on” signal are several triggering receptors, including natural cytotoxicity receptors (NCRs) and natural killer group 2D (NKG2D), whose ligands are mainly stress-inducible molecules. NK cells can attack viral infected and cancer cells that have downregulated HLA class I molecules through “missing self recognition”, and/or have overexpressed ligands of the activating receptors leading to “induced self-recognition”. In peripheral blood (PB), two main NK cell subsets have been identified. A minority is represented by CD56^bright^CD16^−^ NK cells, characterized by the expression of CD94:NKG2A and not KIR, and considered the immature subset. Most PB-NK cells are CD56^dim^CD16^+^ and are extremely diversified in terms of KIRs and CD94:NKG2A phenotype, displaying higher cytotoxic potential [7].

The potent and rapid cytotoxicity exerted by NK cells makes them important and robust effectors in antitumor immunotherapy. NK cells can respond to different types of chemokines released in tumor sites and can release chemotactic high mobility group box 1 (HMGB1) capable of amplifying the antitumor response by attracting additional NK cells at the tumor site [8]. Moreover, preclinical studies and clinical trials have demonstrated the nontoxicity and efficacy of the use of allogeneic NK cells against various hematological malignancies [9,10,11,12]. Although acute myeloid leukemia (AML) patients have been more investigated in NK cell-based approaches, also chronic myeloid leukemia (CML) patients can be considered possible candidates, since recent clinical studies, such as IMMUNOSTIM [13] and EURO-SKI [14], have shown a positive correlation between higher NK cell numbers after imatinib discontinuation and molecular relapse-free survival.

In this review, we first describe the NK cell biology with the various receptor/ligand interactions governing their capability to attack malignant cells, particularly of hematological origin, and then the different immunotherapeutic approaches employing autologous or allogeneic NK cells, in transplantation and non-transplantation setting, either un-activated or potentiated by different systems including cell engineering.

## 2. NK Cell Receptors

### 2.1. HLA-Specific NK Receptors

Two main types of NK cell receptors, capable of recognizing HLA class I molecules, are KIRs and CD94:NKG2 heterodimers, whose expression is mainly confined to NK cells and small subsets of T cells [15]. In addition, leukocyte immunoglobulin like receptor B1 (LILRB1) (also named ILT-2, LIR-1, or CD85j) is not only present on NK and T but also, at high surface density, on B and myeloid cells. LILRB1, interacting with conserved α3 domain and β2 microglobulin, recognizes a broad spectrum of classical and nonclassical HLA class I molecules [16].

KIRs are type I molecules, including both inhibitory (iKIR) and activating (aKIR) receptors [15,17]. Their nomenclature reflects their structure and function: KIR2D and KIR3D indicate two or three extracellular domains, followed by L (long) or S (short), related to the cytoplasmic tail of iKIR or aKIR, respectively [18]. Inhibitory KIRs have a long cytoplasmic tail that contains immunoreceptor tyrosine-based inhibitory motifs (ITIM), able to transduce an inhibitory signal through the recruitment of tyrosine phosphatases. Conversely, aKIRs are characterized by short cytoplasmic tails lacking ITIM motifs and display a positively charged amino acidic residue (Lys) in the transmembrane region, which mediates the association with KARAP/DAP12, a molecule containing immunoreceptor tyrosine-based activating motifs (ITAM) [19,20,21]. An exception is represented by KIR2DL4, a receptor characterized by both a long cytoplasmic tail, including a single ITIM motif, and a charged amino acid (Arg) in the transmembrane region, allowing its association with γ chain of FcεRI. Notably, in resting NK cells, engagement of KIR2DL4 results in strong cytokine (IFN-γ) production [22].

The polygenic and polymorphic *KIR* gene family maps on chromosome 19p13.4 and consists of 13 genes and 2 pseudogenes. *KIR* genes are organized in haplotypes and the two main groups, varying for both type and number of gene content, are termed A and B. Generally, group A haplotypes comprise a fixed number of genes, most of which encoding iKIR (*KIR2DL1*, *KIR2DL3*, *KIR3DL1*, and *KIR3DL2*) with only one aKIR (*KIR2DS4*), and display high degree of allelic polymorphism. Conversely, group B haplotypes are characterized by a greater gene content diversity, including a variable number of aKIR (*KIR2DS1*, *KIR2DS2*, *KIR2DS3*, *KIR2DS4*, *KIR2DS5,* and *KIR3DS1*), and by low allelic polymorphism [23]. A recombination hot spot, located between *KIR3DP1* and *KIR2DL4*, splits the haplotypes into centromeric (Cen) and telomeric (Tel) regions [24]. Various combinations of Cen and Tel regions can be created, and while *KIR* A haplotypes are composed by Cen-A/Tel-A, all the others correspond to *KIR* B haplotypes.

The four main iKIRs are specific for epitopes shared by distinct groups of HLA class I allotypes, also named KIR-ligands (KIR-L). The dimorphism at position 80 defines two mutually exclusive HLA-C epitopes, differentially recognized by KIR2DL1 and KIR2DL2/L3. In particular, KIR2DL1 recognizes HLA-C^K80^ allotypes (HLA-C2 epitope), while KIR2DL2/L3 recognize HLA-C^N80^ allotypes (HLA-C1 epitope). In addition, KIR3DL1 is specific for HLA-B or HLA-A molecules sharing the Bw4 public epitope (Bw4^I80^ or Bw4^T80^), and KIR3DL2 binds HLA-A*03 and -A*11 allotypes [15]. Regarding aKIR, except for KIR2DS1 recognizing HLA-C2 as its inhibitory counterpart (KIR2DL1) [25], their ligand recognition remained elusive for a long time. The actual view of KIR/KIR-L interactions appears more and more complex, taking into consideration the *KIR* allelic polymorphism and the diverse repertoire of peptides bound to the polymorphic HLA class I molecules. Updated data have been recently reviewed [26].

Other receptors are represented by the inhibitory CD94:NKG2A and the activating CD94:NKG2C heterodimers, composed by type II proteins belonging to the C-type lectin superfamily, which recognize the non-classical HLA class I molecule HLA-E [27]. HLA-E is characterized by limited polymorphism and binds a restricted set of peptides, mainly derived from the leader sequences (from −22 to −14 residues) of HLA-A, -B, or -C molecules. For this reason, CD94:NKG2A is considered a sensor of the overall amount of HLA class I expressed on the cell surface. The M/T dimorphism at position −21 of the leader sequence of HLA-B has been described as impacting the CD94:NKG2A/HLA-E interaction. Indeed, in −21M HLA-B individuals, higher HLA-E expression and more efficient NKG2A^+^ NK cells have been detected [28]. This feature has been shown relevant in NK cell activity against AML blasts, which display a low HLA-E expression. Studying a cohort of AML patients receiving histamine dihydrochloride and low dose interleukin (IL)-2 to prevent relapse, Hallner et al. found that patients carrying at least one −21M HLA-B had a better clinical outcome [29].

Among different individuals, a great heterogeneity of PB-NK cell phenotypes, particularly within the CD56^dim^ subset, can be observed. This diversity is primarily due to the high polymorphism of *KIR* and *HLA* class I genes, which segregate independently, leading to diverse compound genotypes [17]. In addition, the clonal distribution of KIRs and CD94:NKG2A, which are epigenetically regulated, creates highly stochastic repertoires of self-tolerant NK cells, following the rules of NK cell “education” [6]. Thus, each competent NK cell, expressing at least an inhibitory receptor specific for self-HLA, can attack unhealthy cells that have downregulated HLA class I molecules through “missing self” recognition. By the same mechanism, NK cells can be alloreactive when expressing only “educated” inhibitory KIR(s) that are not engaged by the HLA class I molecules (i.e., KIR-L) present on allogeneic cells. This situation frequently occurs in HLA-haploidentical hematopoietic stem cell transplantation (haplo-HSCT), when a KIR/KIR-L mismatch in graft-versus-host (GvH) direction is present. Genetically defined by donor *KIR* gene profile and donor/recipient KIR-L, the actual size of the alloreactive NK cell subset can greatly differ among different donors [11].

### 2.2. Non-HLA Specific Activating NK Receptors

One of the earliest functions described for NK cells was the capability to perform antibody dependent cell-mediated cytotoxicity (ADCC) through the engagement of CD16, the low affinity receptor for fragment crystallizable (Fc) of IgG (FcγRIIIa) [30]. This activating receptor can be exploited to potentiate the antitumor NK cell activity in adoptive immunotherapy by the use of IgG antibodies recognizing tumor-associated antigens, bispecific, or trispecific killer engagers (BiKEs and TriKEs, respectively), as described in 4.2 paragraph.

In addition, NK cells express a set of triggering receptors and coreceptors that deliver the “on” signal upon interaction with specific ligands on tumor target cells. Major NK cell receptors are the natural cytotoxicity receptors (NCRs) [31], type I molecules of the Ig family, consisting of three members: NKp46, NKp30, and NKp44 [32,33,34,35,36]. While NKp46 and NKp30 are expressed on virtually all resting NK cells, NKp44 is acquired upon activation. NCR transmembrane domains contain a positively charged amino acid, allowing the association with ITAM-bearing adaptor proteins, namely FcεRIγ and/or CD3ζ for NKp46 and NKp30 [31], as well as DAP12 for NKp44 [37]. Although NCR expression was considered confined to NK cells, more recently, one or another NCR has been also detected on subsets of ILC [2,38,39]. Notably, upon appropriate culture conditions, NKp30 is inducible on γδT cells or CD8^+^ αβT cells that acquire a “gain of function”, and thus, an enhanced leukemia recognition and killing [40,41,42]. Functional evidences that NCRs, especially NKp46, play a primary role in leukemia recognition and killing induction have been provided [43]. Although many NCR ligands (NCR-Ls) have been characterized, the panel of the membrane bound molecules, possibly overexpressed in hematological malignancies, appears incomplete [44,45]. In this regard, relevant surface molecules are B7-H6 and a splice variant of mixed-lineage leukemia 5 (21spe-MLL5), identified as NKp30-L and NKp44-L, respectively [46,47]. Importantly, the interaction between NKp44 and a subset of HLA-DP molecules (i.e., HLA-DP401) has been recently proven to trigger functional NK cell responses [48]. Other NCR-Ls are represented by nuclear antigens that can reach the plasma membrane during tumor transformation and can be expressed in exosomes by tumor cells: Proliferating cell nuclear antigen (PCNA), recognized by NKp44, and HLA-B-associated transcript 3 (BAT3), also known as BCL2-associated athanogene 6 (BAG6), by NKp30 [49,50]. Several soluble NCR-Ls have been identified, including complement factor P (CFP) recognized by NKp46, platelet-derived growth factor (PDGF)-DD and nidogen-1 by NKp44, in addition to soluble forms of the NKp30-L BAG6 and B7-H6 [51,52,53,54]. Soluble NCR-Ls are studied as biomarkers for cancer patients. Regarding hematological malignancies, in chronic lymphocytic leukemia (CLL) patients, high plasma levels of soluble BAG6 have been associated with advanced disease stages; in contrast, NK cells were activated by BAG6 presented on the surface of exosomes [55]. The shedding of NCR-Ls by neoplastic cells can be envisaged as a tumor immune escape mechanism; indeed, it can induce receptor surface downmodulation with consequent NK cell dysfunction. The observation of NK cells with NCR^dull^ phenotype [56] has been described in AML and CLL patients [57,58,59]. Moreover, impairment in NCR expression and function can be induced by hypoxia or additional soluble factors present in the tumor microenvironment, including indoleamine 2,3-dioxygenase (IDO), transforming growth factor-beta (TGF-β), and prostaglandin E2 (PGE2) [60,61,62]. Indeed, in patients affected by solid and hematologic tumors, NCR^dull^ NK cells can be detected in peripheral blood but particularly in the tumor site [63]. Moreover, TGF-β contributes in the plasticity of group 1 ILCs, driving the conversion of NK cells into ILC1. This represents a mechanism of immune evasion in the tumor microenvironment [64].

Another major NK-activating receptor is the NKG2D homodimer, a type II and C-type lectin-like molecule, which is expressed on all NK and cytotoxic T lymphocytes, mainly γδT cells and CD8^+^ αβT cells. Upon receptor engagement, human NKG2D transduces an activation signal via the associated transmembrane adaptor protein DAP10. Multiple NKG2D ligands (NKG2D-Ls) have been characterized and are represented by MHC class I chain-related protein A/B (MICA/B) and UL16 binding proteins (ULBP)1-6 [65]. It’s well known that their expression on the cell membrane is induced upon stress and malignant transformation. MICA/B are transmembrane molecules and have been mainly described on epithelial tumors and melanoma [66,67]. ULBP1/3/6 are GPI-anchored molecules and ULBP4/5 are transmembrane molecules, whereas ULBP2 can be in both forms. NKG2D-L expression on primary leukemias and the consequent involvement of NKG2D in NK cell-mediated target recognition has been documented [43,68,69]. In leukemia patients, the levels of NKG2D-Ls, as surface expressed and/or in soluble form, have been correlated with NKG2D downregulation and reduced NK cell function and clinical data, underlying the relevance of NKG2D-mediated tumor immunosurveillance and escape [68].

NK cells are also equipped with costimulatory receptors, which can collaborate with NCR and NKG2D, enhancing the activating signaling and NK cell function. They include DNAM-1 [70], 2B4 (CD244) [71], NTB-A [72], CD59 [73], and NKp80 [74]. Relevant for antileukemia activity, DNAX accessory molecule 1 (DNAM-1) (CD226) can recognize the specific ligands poliovirus receptor (PVR) (CD155) and Nectin-2 (CD112), found to be expressed on various acute leukemias [43,75]. Belonging to the signaling lymphocytic activation molecule (SLAM) family receptors, NK-T-B-antigen (NTB-A) displays homophilic interaction like the other members, while 2B4 recognizes CD48, exclusively present on hematopoietic cells. Upon receptor engagement, the ITSMs in their cytoplasmic tail become phosphorylated and associate with SLAM associated protein (SAP), which in turn activates downstream signaling pathways resulting in NK cell activation. High levels of CD48 and NTB-A are expressed by Epstein-Barr virus (EBV) infected B cells and lymphomas [76]. Conversely, a downregulation of these molecules is often observed on acute leukemia cells [43].

In addition, NK cells can be activated by the recognition of bacterial or viral products via toll-like receptors (TLRs) [77,78].

### 2.3. Inhibitory Checkpoints Expressed on Human NK Cells

In addition to the HLA class I specific inhibitory receptors (iKIRs and NKG2A), additional inhibitory checkpoints, responsible for maintaining the immune cell homeostasis, can be expressed on human NK cells. They include programmed death-1 (PD-1), T-cell Ig and ITIM domains (TIGIT), CD96, and T-cell Ig and mucin domain-containing protein 3 (TIM-3) [79,80]. In pathological conditions, such as hematological malignancies, high expression of ligands for inhibitory checkpoints have been associated with poor prognosis [81]. Indeed, the tumor microenvironment can induce the de novo expression of some of these immune checkpoints on tumor-associated NK cells, thus facilitating tumor immune escape.

PD-1 (CD279 or PDCD1) is a major checkpoint expressed by NK cells [82,83]. It binds to PD-L1 (CD274, B7-H1, or PDCD1LG1) or PD-L2 (CD273, B7-DC, or PDCD1LG2), with the highest affinity for PD-L2. PD-L1 expression is usually low on healthy tissues [84], but is upregulated on various tumor types upon exposure to inflammatory conditions (e.g., IFN-γ) or following activation of key oncogenic pathways involving phosphoinositide 3-kinase (PI3K) or mitogen-activated protein kinase (MAPK). On the other hand, PD-L2 is expressed by antigen presenting cells and by certain solid tumors. The molecular mechanisms regulating the expression of PD-1 on human NK cells have not been defined so far. However, it is conceivable that signals delivered by cells and/or soluble factors present in the tumor microenvironment may play an important role [85].

The TIGIT and CD96/Tactile immune checkpoints [86,87] compete with the activating receptor DNAM-1 for binding to PVR and Nectin-2, molecules that are usually upregulated in tumor cells [75]. A recent report has suggested that TIGIT targeting with specific mAbs may unleash T and NK cell antitumor activity and prevent NK cell exhaustion [88].

Other inhibitory checkpoints that may be expressed by NK cells are lymphocyte-activation gene 3 (LAG-3) and TIM-3. While blockade of TIM-3 has been shown to increase NK cell cytotoxicity in preclinical models [89], the effect of LAG-3 on NK cell function is still unclear and requires further investigation [90,91]. The main ligand of TIM-3 is galectin-9 [92], but other ligands have been identified, such as phospatidyl serine (PtdSer) [93], HMGB1 [94], and carcinoembryonic antigen-related cell adhesion molecule 1 (CEACAM1) [95].

NKRP1A (CD161), a C-type lectin-like inhibitory receptor that recognizes lectin-like transcript 1 (LLT1), can be considered a putative checkpoint receptor. Indeed, since LLT1 is expressed by different tumors, including B-cell non-Hodgkin’s lymphomas (NHLs) [96], this receptor/ligand pair may play a role in tumor escape from NK cell control. Moreover, blocking NKRP1A/LLT1 interaction increases NK cell-mediated secretion of IFN-γ and killing of NHL cell lines. Thus, the use of anti-LLT1 blocking mAbs may improve tumor immunosurveillance and also enhance the efficacy of anti-CD20-based immunotherapy strategies [97]. Immunoregulatory cytokines in the microenvironment can modulate NKRP1A expression and consequently NK cell function. IL-12 induces upregulation of NKRP1A in NK cells, leading to a strong inhibition of the cytolytic activity mediated by CD16 or NKp46 [98]. Conversely, IL-2 reduces NKRP1A expression in NK cells, possibly contributing to enhanced killing activity [99].

## 3. NK Cells in Haploidentical HSCT and Adoptive Immunotherapy

### 3.1. T Cell-Depleted and T Cell-Replete HSCT

Allogeneic HSCT is a life-saving treatment for patients affected by high-risk malignant hematologic disorders. However, only 25% of patients who need an allograft have an HLA-identical sibling available as donor. Thus, alternative donors and sources of hematopoietic stem cells (HSC) can be matched to unrelated volunteers, unrelated umbilical cord blood (UCB), and HLA-haploidentical relatives (i.e., a family member sharing one HLA-haplotype with the recipient) [100]. Haplo-HSCT offers an immediate option to almost all patients in need of an allograft. However, because of multiple HLA class I and II disparities between donor and recipient, bidirectional alloreactivity to incompatible HLA molecules can cause important clinical complications, including graft failure and the incidence of both acute and chronic graft-vs-host disease (GvHD). Donor-derived T cells are the most responsible for the occurrence of severe GvHD, and different T-cell depletion strategies or pharmacological immunosuppressive treatments have been employed [9,10,11].

Pioneering clinical studies by the Perugia group rendered successful haplo-HSCT, through the use of intense conditioning regimen preventing graft rejection, and inoculum of “megadoses” of highly purified CD34^+^ cells, thus avoiding GvHD [101]. Ruggeri et al. demonstrated, in adult AML patients, an efficient post-transplant NK cell recovery and protective graft-vs-leukemia (GvL) effects mediated by alloreactive NK cells, in the absence of GvHD. Indeed, transplantation from donors characterized by alloreactive NK cells was associated with a lower incidence of relapse and an improvement of the overall survival in AML adult patients [102,103,104]. The beneficial effect of donor NK alloreactivity was also observed in children with high-risk acute lymphoid leukemia (ALL) showing 70% versus 35% survival rate in the presence versus absence of NK alloreactivity, respectively [11]. The first lymphocyte population that reconstitutes after HSCT is represented by NK cells displaying an immature phenotype, and several months are necessary for their acquisition of full phenotypic and functional maturation [105]. In pediatric leukemia patients undergoing T cell-depleted haplo-HSCT, donor-derived alloreactive NK cells displaying antileukemia activity were generated, appeared in PB after 2–3 months, and could persist for years in the recipient. Moreover, a positive role of KIR2DS1^+^ NK cells derived from HLA-C1/C2 donors was also demonstrated in the recognition and killing of HLA-C2/C2 leukemia cells [106]. However, a general problem occurring in T cell-depleted haplo-HSCT is the delayed immune recovery, increasing the risk for patients of life-threatening opportunistic infections [100,104].

In recent years, a T cell–replete haplo-HSCT was developed by the use of an unmanipulated graft and post-transplant high-dose cyclophosphamide (PTCy) administration, followed by other immunosuppressive drugs, to prevent GvHD. The use of PTCy aims to selectively eliminate alloreactive T cells rapidly proliferating in response to the recipient alloantigens. Thus, in this transplantation setting, an accelerate immune reconstitution by the maintenance of a broad repertoire of nonalloreactive T lymphocytes, potentially active against post-transplant infections, might be achieved [107]. Morever, Russo et al. reported that donor-derived NK cells, proliferating for the high systemic levels of IL-15, become sensitive to Cy-mediated killing. Thus, early elimination of all mature NK cells, including the alloreactive subset, has been documented. The delayed recovery of mature NK cells through the differentiation from precursors might result in an impaired NK-mediated antileukemic potential. Consequently, no evidence of a beneficial effect of donor NK alloreactivity on the outcome of patients was observed [108]. In addition, in a retrospective multicenter analysis, KIR-L mismatching was associated with a worse outcome in leukemia patients receiving haplo-HSCT and PTCy when peripheral blood stem cells (PBSC) were used as graft cell source [109].

### 3.2. Adoptive NK Cell Immunotherapy within Transplant Setting

In both CD34^+^ and PTCy haplo-HSCT, the delayed recovery of mature NK cells can impair their GvL effect and protection against viral infections. To circumvent this problem, adoptive immunotherapy approaches with highly purified donor-derived NK cells have been largely investigated in different transplantation settings. Several clinical trials are currently active and ongoing. Although the efficacy appeared limited to a minority of patients, NK-donor lymphocyte infusions (NK-DLI) prior to or post-HSCT were shown to be feasible without severe side effects [110,111,112].

In a pilot study on AML patients, donor NK cells were infused after haplo-HSCT, to consolidate incomplete engraftment [113]. Indeed, purified NK cells, preferentially recognizing hematopoietic host cells, promoted engraftment without inducing GvHD. The safety and feasibility of the adoptive transfer of allogeneic NK cells were further confirmed in a phase II study, but NK-DLI had no apparent effect on graft failure or relapse incidence [114].

In a phase I/II trial, in which AML patients underwent haplo-HSCT in combination with early transfer of NK cells, a two year overall survival rate of 37% was observed, suggesting that adoptively transferred NK cells possibly contribute to long-term remission in patients with refractory AML [115].

The favorable cytokine environment, with high systemic levels of IL-15, in patients immediately after haplo-HSCT with PTCy [108], provided the rationale for NK-DLI to reduce relapse incidence and viral infections. Clinical trials based on the infusion of donor NK cells after haplo-HSCT and PTCy have been started [116,117]. In a phase I study for high-risk myeloid malignancies, high doses of ex-vivo expanded donor-derived NK cells infusions significantly improved NK cell function and may be effective to prevent leukemia relapse with no major toxicity [117].

Notably, it should be considered that, in addition to the characteristics of the different transplantation settings, several NK cell-related variables, including their preparation (either unstimulated or ex-vivo activated/expanded), dose, timing of the infusion, and presence of NK cell alloreactivity, may also influence clinical responses [118,119].

### 3.3. Haploidentical HSCT after CD3/CD19 or TCRαβ/CD19 Depletion

New graft manipulations have been employed with the aim to infuse NK cells together with HSC [120]. In haplo-HSCT, depletion of CD3^+^CD19^+^ cells, instead of CD34^+^ cell selection, leads to a better engraftment and immune reconstitution [121,122]. This procedure allows the elimination of T cells, responsible for GvHD, and CD19^+^ B cells, to prevent post-transplant EBV-related lymphoproliferative disorders. This graft manipulation has been applied for both children and adult acute leukemia patients. Although promising clinical results have been reported, GvHD incidence was quite high due to different levels of T cell depletion, requiring the development of more efficient procedures. A further improvement has been represented by the approach based on selective TCR αβ T and CD19 B cell depletion from mobilized PBSC [123,124]. This refined procedure allows the accurate removal of αβT cells, responsible for GvHD, and the infusion of a graft enriched for HSC and also containing other cell types, including mature NK cells and γδT lymphocytes (NCT01810120). The presence of these mature effectors can favor the engraftment and reduce the risk of infections and leukemia recurrence. Thus, in αβT and B cell-depleted haplo-HSCT, high numbers (about 20–40 millions/kg of recipient body weight) of donor mature NK cells, including alloreactive populations, are immediately available and can fully display their activity because of the absence of pharmacological GvHD prophylaxis. NK cells may promptly exert their antileukemia and GvHD-preventing effects in the 6–8 weeks after transplantation, before the emergence of KIR^+^ NK cells differentiated from CD34^+^ precursors. Transplanted patients also benefit from many γδT cells, which contribute to anti-infectious and possibly to antileukemia activities [125]. Importantly, the clinical outcomes of pediatric leukemia patients receiving αβ T and B cell-depleted haplo-HSCT were very good, showing high leukemia-free survival (LFS, 71% and 68% in high risk ALL and AML, respectively) and low risk of GvHD [126]. In this cohort, the donor (either mother or father of the patient) was mainly chosen according to immunological criteria, giving priority to NK alloreactivity, *KIR* B/x genotype, higher B-content score, and larger size of alloreactive NK cell subset [106,127,128,129] (Figure 1). In addition, donor/recipient HCMV serology, donor age, donor/recipient body weight, presence of *KIR2DS1* in HLA-C1^+^ donor and HLA-C2^+^ recipients, higher percentage of NK and γδT lymphocytes, higher expression of NKp46, and presence of NKG2C were also evaluated [11,43,106,130]. Recently, the European Society for Blood and Marrow Transplantation (EBMT) elaborated consensus recommendations for donor selection in haplo-HSCT [131].

### 3.4. Adoptive NK Cell Immunotherapy in Nontransplant Setting

Antileukemia activity of the adoptive transfer of NK cells from haploidentical donors in nontransplant settings has been also explored. In the first study by Miller et al., 19 adult AML patients, undergoing different preparative regimens, were infused with overnight IL-2 activated haploidentical NK cells followed by daily subcutaneous injection of IL-2 for 14 days [132]. Adoptively transferred human NK cells were safe and could be expanded in vivo. Indeed, circulating haploidentical NK cells were observed up to 28 days after infusion. The expansion was observed in patients with the more intense Cy/Flu preparative regimen, which was associated with high serum concentrations of IL-15. Notably, 5 of 19 poor prognosis AML patients achieved complete remission (CR) after haploidentical NK cell therapy, with a significantly higher rate when KIR ligand mismatched donors were used. IL-15 could be a suitable alternative of IL-2 for adoptive NK cell therapy because it avoids Treg stimulation (see 4.1 paragraph). The results of the first phase I/II clinical trials with recombinant human IL-15 (rhIL-15), administered either by subcutaneous (sc) or intravenous (iv) injection, and haploidentical NK cell therapy after lymphodepletion in relapsed/refractory AML patients have been recently published [133]. Beneficial clinical responses have been observed, and rhIL-15 induced in vivo NK cell expansion and remission rates better than those observed in previous trials with IL-2. However, unexpectedly, after sc and not iv treatment with rhIL-15, high frequency of cytokine release syndrome (CRS) and neurotoxicity was observed. Further studies of pharmacokinetics and pharmacodynamics will be necessary to optimize the therapeutic benefits of IL-15 and minimize CRS.

Additional studies have shown the efficacy of the adoptive transfer of haploidentical KIR/KIR-L mismatched NK cells in children, adults, and high-risk elderly patients with AML, not candidates to receive HSCT. In the pilot study of haploidentical NK cell transplantation for AML (NKAML), the safety and feasibility of low-dose immunosuppression followed by the infusion of highly purified haploidentical NK cells in children with AML were assessed [134]. All patients showed safe engraftment, with an expansion of donor-derived alloreactive NK cells during the first four weeks after infusion, and remained in CR after a follow-up of approximately 32 months.

In high-risk elderly patients with AML infused with highly purified alloreactive NK cells from haploidentical donors after Cy/Flu immunosuppressive chemotherapy followed by in-vivo IL-2 administration, no NK cell-related toxicity, including GvHD, was observed [135]. The procedure was beneficial for the outcome of patients treated in CR or very early in molecular relapse, or both. Thus, the infusion of purified NK cells is feasible in elderly AML patients as a post-CR consolidation strategy. This study also documented the kinetics of emergence and persistence over time of functional donor-versus-recipient NK cell alloreactivity after NK cell infusion. In addition, the infusion of high numbers of alloreactive NK cells can improve the clinical efficacy of adoptively transferred haploidentical NK cells [136].

Therefore, the infusion of a defined number of functionally active NK cells could be of great impact on the efficacy of NK cell-based treatment. The selection of haploidentical donors predictive of patients response upon adoptive transfer of NK cells should be valorized, as well [137].

## 4. Strategies to Induce NK Cell Activation, Persistence, and Expansion

### 4.1. Cytokine-Mediated NK cell Activation and Expansion to Improve Tumor Killing

Antitumor NK cell functions can be modulated not only by the direct ligand–receptor interactions, but also by several cytokines. In particular, IL-2, IL-12, IL-15, IL-18, and IL-21 can activate NK cells, whereas suppressive cytokines, including TGF-β or IL-10, can inhibit them [138,139]. To improve antitumor responses, cytokine therapies capable of supporting NK cell differentiation, activation, persistence, and expansion have been tested in preclinical studies and clinical trials [140,141] (Figure 2).

The first cytokine shown to exert a relevant role in the treatment of tumors was IL-2 [142]. IL-2 can mediate its effects by binding to a high-affinity receptor composed of IL-2Rα, IL-2Rβ, and the common γ chain. In NK cells the engagement of the IL-2R complex by IL-2 leads to phosphorylation of STAT-1, -3, and -5; activation of p38 MAPK [143]; and later, induction of NK cell proliferation and increment of NK cell cytotoxicity [144]. However, the administration of high doses of IL-2 is associated with an increased risk of severe adverse reactions, including vascular leakage and organ injury caused by activation of the vascular endothelium (which is characterized by the expression of the IL-2 high affinity receptor, IL-2Rαβγ), and also with poor therapeutic index [145,146,147,148,149,150]. Indeed, IL-2 can also bind to the IL-2Rαβγ on Treg cells and, as a consequence, it can indirectly inhibit NK cell proliferation and cytotoxicity through the action of TGFβ released by IL-2 activated Treg cells [151], and through the deprivation of IL-2 by Treg cells [152,153]. On the other hand, it is important to remember that the use of low doses of IL-2 to expand NK cells after autologous transplantation was shown to result in efficient in vivo expansion of NK cells [154], but with limited antitumor efficacy, probably due to inhibitory signals occurring upon KIR/NKG2A-mediated self-HLA class I engagement and the stimulation of suppressive Treg cells by IL-2 [155,156]. Thus, high doses of IL-2 were necessary for an efficient antitumor activity, but were also responsible for inducing severe toxicity. For these reasons, strategies useful for the production of modified forms of IL-2, capable of immunoactivation and avoiding immunosuppression, became necessary. Along this line, mutant forms of IL-2 with high affinity for IL-2Rβγ present on NK cells, but reduced affinity for IL-2Rα expressed on Treg cells, have been generated [157,158,159] (Figure 2).

Another cytokine capable of activating NK cells is represented by IL-15, a soluble factor that shares many activities with IL-2 (probably because these cytokines use common receptor subunits), but that, differently from IL-2, does not induce Treg-mediated immune suppression [160]. IL-15 interacts with a heterotrimeric receptor composed by the β and common γ chain of the IL-2R [161] and the unique high affinity IL-15-binding subunit, called IL-15Rα [162,163]. Notably, cells expressing IL-2Rβγ, but not IL-15Rα, can bind and respond to IL-15 only when high concentrations of this cytokine are present. On the contrary, IL-15Rα binds to IL-15 with high affinity, also in the absence of the IL-2Rβγ. Moreover, on the surface of dendritic cells or other myeloid cells, IL-15Rα can present IL-15 in trans to IL-2Rβγ receptors expressed on NK and CD8^+^ T cells, without activating Tregs [164]. However, the clinical use of IL-15 is impaired by its short half-life. Strategies to improve IL-15 administration and dosing are still being studied to optimize its biological effects, reducing toxicity [133]. Rubinstein and colleagues demonstrated that the combination of IL-15 with the IL-15Rα subunit formed a soluble compound (termed IL-15 superagonist) with significantly longer half-life and higher biological activity than native IL-15 [165].

In order to further increase the in vivo half-life of IL-15, the IL-15 superagonist ALT-803 has been recently developed by binding an IL-15 mutant (IL-15N72D) to a soluble, dimeric IL-15Rα Fc fusion protein (IL-15Rα-Fc) (Figure 2). This compound has a prolonged half-life and an increased ability to bind IL-2Rβγ and mediate immunostimulatory functions as compared to IL-15 alone [166,167]. Preclinical studies have shown that ALT-803 can mobilize both innate and adaptive immune responses by enhancing NK and T cell functions. Recently, in a phase I study, clinical benefits upon iv or sc ALT-803 administration have been observed in patients with hematologic malignancies who had relapsed after allogeneic HSCT [168]. In these patients, ALT-803 was generally well-tolerated, with no severe toxicities and GvHD. Moreover, ALT-803 has been used as a functional scaffold for creating multispecific, targeted IL-15-based immunotherapeutic agents to enhance tumor clearance. Indeed, ALT-803 has been fused to four single chains of Rituximab to generate the 2B8T2M molecule, a compound displaying trispecific activity: recognition of CD20 on tumor cells, stimulation of IL-2Rβγ on immune cells, and binding of FcγR on NK cells and macrophages [169]. Thus, NK cells can be activated inducing the killing of B-lymphoma cells through ADCC.

A recent study has demonstrated that the IL-15-AKT-XBP1s signaling pathway contributes to enhance antitumor effector functions and NK cell survival. In particular, the protein stability of XBP1s, induced by the IL-15-mediated phosphorylation of AKT, positively regulates the expression of granzyme B and the antileukemia activity of NK cells [170].

The short ex vivo treatment of NK cells with a combination of IL-15, IL-12, and IL-18 before NK cell adoptive transfer is another promising cytokine-based approach for antitumor clinical applications (Figure 2). Indeed, these cytokine-primed NK cells, called “cytokine induced memory-like NK cells” (CIML-NK), have been shown to be long-lived and memory-like NK cells, characterized by enhanced IFN-γ production and cytotoxicity against tumor cells [171,172,173,174]. Preactivation of NK cells with IL-18/IL-15/IL-12 was also shown to increase the expression of CD25 and, as a consequence, favor the survival of NK cells in patients, even without administration of exogenous IL-2 and any toxicity. On the other hand, IL-18/IL-15/IL-12 treatment could also induce negative effects on NK cells, for example the reduction of CD16 expression that, however, could be restored by removing cytokine stimulation. Notably, this type of NK cell expansion allowed large-scale NK cell production, useful for repeated therapeutic use, and the adoptive transfer of these cytokine-primed NK cells was effective, as demonstrated in murine cancer models and in clinical trials [171,172]. CIML-NK cells display enhanced IFN-γ production and cytotoxicity against leukemia cell lines or primary human AML blasts in vitro, regardless of KIR/KIR-L interactions. Moreover, a first-in-human phase I clinical trial demonstrated CIML-NK cell expansion and robust responses against AML blasts [175].

Another cytokine, which is known to be involved in development/proliferation of NK cells from progenitor cells, induction of NK cell receptor expression, IFN-γ secretion, and cytotoxicity, is IL-21. However, it is important to underline that the role of this cytokine on NK cell function is controversial; indeed, it has also been reported to trigger apoptosis and diminish the positive effects of IL-15 [176]. According to the high potential of IL-15 in NK cell expansion and the effects of IL-21 on NK cell maturation and function [177,178], a two-phase expansion protocol based on the use of IL-15 to induce an early NK cell expansion, followed by short exposure to IL-21 to boost NK cell cytotoxicity against tumor cells, has been developed [179].

Moreover, a method to expand NK cells ex-vivo using genetically modified K562 feeder cells equipped with membrane-bound IL-21 (K562mb-IL-21) has been developed [180]. Recently, this method, used in a phase I clinical trial for the ex vivo expansion of donor-derived NK cells in haplo-HSCT, has been demonstrated to be safe and effective in controlling leukemia with no major toxicity and to be associated with significantly improved NK cell number and function, lower viral infections, and low post-transplant relapse rate [117].

### 4.2. CD16-Mediated Tumor Cell Killing to Cure Hematological Malignancies

Besides the use of cytokines to drive NK cell activation and function against malignancies, other immunotherapeutic strategies enhancing NK cell antitumor potential are based on the innate ability of NK cells to kill target cells opsonized with antibodies via ADCC. This mechanism implies the engagement of the activating receptor CD16 (FcγRIIIa), which recognizes and binds the immunoglobulin Fc fragment with low affinity [30]. The first therapeutic interventions taking advantage of CD16 function on NK cells were based on the administration of tumor-targeting chimeric monoclonal antibodies (mAbs), such as rituximab, a mAb recognizing CD20 that still represents a first-line treatment in B-chronic lymphocytic leukemia (B-CLL) [181] (Figure 2). To increase mAbs’ affinity for CD16, humanized mAbs, such as obinutuzimab (anti-CD20), have also been generated by engineering the Fc fragment [182], possibly translating their higher affinity to a better clinical outcome [183]. Interestingly, these modifications were capable of augmenting CD16 affinity for IgG also in individuals carrying the polymorphism that decreases CD16 Fc-binding capacity (i.e., bearing a CD16A-158F allotype instead of the high affinity CD16A-158V allotype) [184,185,186]. Indeed, the low-affinity CD16 form has been associated with inferior therapeutic effects of rituximab in lymphoma patients [187,188]. Along this line, a Fc-modified anti-CD133 with higher affinity for Fc was shown to elicit improved NK cell responses in a xenograft model of human AML [189].

In view of the high clinical potential of CD16-mediated tumor cell killing to cure malignancies, different approaches have been recently developed to further improve NK cell activation through this receptor. Following the strategy applied to generate T cell engaging antibodies (BiTE), such as blinatumomab, a CD19/CD3-bispecific single chain T-cell engager employed for relapsed/refractory ALL [190,191], bispecific antibodies and BiKEs-triggering NK cells have been produced. While BiTEs can show adverse side effects [192,193], BiKEs promise to be safer, more efficacious, and flexible. These molecules couple immune cell engagement to tumor targeting by forming an immunological synapse between NK and tumor cells. Indeed, BiKEs are composed of a single-chain variable fragment (scFv) of an antibody specific for a given tumor antigen, connected through a short peptide linker to an anti-CD16 scFv, which triggers stronger cytotoxic signals in NK cells as compared to those elicited by Fc fragments binding to CD16 [194]. BiKEs engaging CD16 and recognizing CD19 [195] (Figure 2) or CD33 [196] have been developed and tested also in combination with an inhibitor of the metalloprotease ADAM17 [197] to avoid/limit CD16 shedding from NK cell surface. These novel immune engagers offer high flexibility and can be tailored to better fit clinical needs, such as improvement of NK cell survival and proliferation. To this end, a TriKE incorporating IL-15 has been designed. In particular, the 16 × 15 × 33 TriKE (Figure 2) has shown enhanced NK-mediated killing of AML and MDS in both in vitro and in vivo preclinical models [198,199]. Similarly, a novel CD19-targeting 16 × 15 × 19 TriKE holds great potential to cure refractory B-CLL [200]. In addition, BiKEs and TriKEs can be also engineered to contain two different tumor specificities permitting them to circumvent the complication represented by the emergence of tumor cells lacking the selected tumor antigen. In this context, a TriKE containing anti-CD19 and anti-CD22 has been designed [195] to overcome the possible appearance of CD19^−^ leukemic blasts that was observed upon blinatumomab treatment in around 20% of pediatric B-ALL patients given the drug [201] (Figure 2). Alternatively, more selective tumor antigens have been introduced in TriKE platforms such as C-type lectin domain family 12 member A (CLEC12A), which is highly expressed also on CD33^−^ AML cells and could better contribute to myeloid leukemia targeting by NK cells [202].

Further improvements will be achieved by the use of novel platforms that are under development by different groups and companies. For example, ROCK^®^ (redirected optimized cell killing) is a registered trademark multispecific platform that permits researchers to create tetravalent NK cell engagers composed of a specific CD16A antibody linked to a bispecific anti-tumor antigen [203]. These molecules promise to be efficacious independently of CD16A allotype, to prevent NK cell fratricide, and to avoid inhibition by serum IgG. Indeed, a tetravalent anti-CD30/CD16A tandem diabody (AFM13) has been successfully tested in a phase I trial and a phase II study is planned for relapsed/refractory Hodgkin’s lymphoma patients [204].

Remarkably, the use of immune engagers retargeting and potentiating ADCC could be particularly efficient in patients characterized by the presence of HCMV-driven adaptive NK cells. This peculiar NK cell subset is present at variable proportions in HCMV^+^ healthy donors [205,206] and can develop in noticeable amounts in leukemic patients undergoing HCMV reactivation after HSCT [192,207,208,209]. Adaptive NK cells are usually characterized by a NKG2C^+^CD57^+^ surface signature, epigenetic modifications, and altered signaling molecules expression that enhance their ADCC potential, suggesting that these cells could provide optimal responses to CD16-engaging molecules [210,211]. Besides enhanced ADCC, adaptive NK cells show strong cytotoxicity in response to NKG2C triggering. Along this line, a novel TriKE composed of an anti-NKG2C combined with an anti-CD33 and IL-15 has been successfully used in vitro to augment AML killing by iPSC-derived NK cells (see 5. paragraph) engineered to express NKG2C [212].

Notably, CMV-induced adaptive NK cells could play an inherent role in preventing leukemia relapse and promoting better clinical outcomes, as recently suggested in the HSCT setting [108,208,209]. Interestingly, a protocol aimed at expanding CMV-induced NKG2C^+^ NK cells for cell therapy has been recently developed to treat pediatric T- and precursor B-ALL [213]. The optimization of expansion protocols combined with the use of appropriate immune engagers will fully exploit the antileukemic properties of adaptive NKG2C^+^ NK cells.

Along with CD16 triggering, the engagement of NCRs could be relevant to achieve optimal NK cell activation against acute leukemia and other hematological malignancies. In a very recent study, trifunctional natural killer cell engagers (NKCEs) targeting CD16 and NKp46 combined with an antitumor antigen (e.g., CD20) (Figure 2) have been proven to induce full activation and enhanced target cell killing, as compared to standard mAbs (e.g., rituximab), in both in vitro and mouse models [214].

NK cell-based immune engagers represent a very plastic tool that can be managed and adapted to different patient needs more easily than other approaches (e.g., adoptive cell transfer, engineering). Although clinical trials based on NK cell immune engagers are at the beginning and novel molecules are in early developmental stages, NK cell engagers hold great potential to transform future antileukemic therapies, especially in combination with both conventional chemotherapy or allo-HSCT and other innovative immunotherapeutic strategies, such as cytokine-based stimulation and immune checkpoint inhibitors.

### 4.3. Restoration of NK-Mediated Antitumor Responses by the Use of Antibodies Blocking Immune Checkpoints

A promising therapeutic approach to cure leukemia patients is represented by the use of monoclonal antibodies capable of both disrupting the interactions between the immune checkpoints (expressed on NK cells) and their ligands (expressed on tumor cells), and restoring efficient NK-mediated antitumor responses. For example, the fully human IgG4 mAb lirilumab, directed against a common epitope shared by KIR2D, has been shown to block the KIR/KIR-L interaction and increase NK cell-mediated killing of AML blasts both in vitro and in vivo [215]. Lirilumab showed acceptable safety without significant toxicity in AML and CLL patients [216]. Moreover, although a single-agent phase I trial with lirilumab has not shown significant efficacy in relapsed/refractory MM [217], the lirilumab/lenalidomide combined therapy has displayed a good response in a following phase I clinical trial in patients affected by the same malignancy [218], confirming that combined blockade of different immune checkpoints is a promising therapeutic strategy. Along this line, analyses evaluating the efficacy of lirilumab in combination with other therapeutics are ongoing. For example, there are: (1) a phase II study evaluating the combination of lirilumab with rituximab (anti-CD20 mAb) for relapsed, refractory, or high-risk untreated patients with CLL (NCT02481297); (2) a phase II study evaluating lirilumab in combination with 5-azacytidine for the treatment of patients with refractory/relapsed AML (NCT02399917); (3) a phase II study analyzing the combined use of lirilumab and nivolumab with 5-azacitidine in patients with myelodysplastic syndromes (MDS) (NCT02599649).

Another mAb used in immunotherapy to potentiate NK cell function is represented by the humanized anti-NKG2A monalizumab (IPH2201), which is capable of blocking the NKG2A/HLA-E interaction [219]. Various clinical trials are evaluating the efficacy of monalizumab in different types of tumors. Regarding the treatment of hematological malignancies, a phase I clinical trial based on the use of monalizumab as monotherapy is ongoing to determine the safety of monalizumab after HLA-matched allogenic HSCT (NCT02921685). Moreover, a phase I/II clinical trial evaluating the combined use of monalizumab with the Bruton’s tyrosine kinase inhibitor ibrutinib in patients with relapsed, refractory, or previously untreated CLL is ongoing (NCT02557516).

## 5. Adoptive Cell Therapy Using Chimeric Antigen Receptor-Engineered Natural Killer Cells

In lymphoid B-cell neoplasia, adoptive cell therapy based on the use of T cells engineered with a chimeric antigen receptor (CAR)-T has achieved exciting results [220,221,222]. Two anti-CD19 CAR-T therapies have been approved by the Food and Drug Administration (FDA) first, and then by the European Medicine Agency, for treatment of relapsed B-cell ALL and refractory/relapsed large B-cell non-Hodgkin’s lymphoma (NHL). However, the application of CAR-T cells is hampered by several obstacles, limiting a widespread clinical use; they include: (i) high cost associated with the drug product, since it requires a single manufacturing for each patient; (ii) significant delay between patient enrollment and treatment, this being associated with the need to define salvage therapy between apheresis and CAR-T cell infusion; (iii) the lack of possibility to reinfusing the drug product in patients experiencing low CAR-T cell persistence or disease relapse; (iv) the majority of patients experience high rate of toxicity, due to the production of IFN-γ and the consequent induction of the CRS and/or neurotoxicity; (v) difficulties in the management of the industrial chain for the autologous drug product’s worldwide distribution.

In this scenario, the development of an allogeneic platform based on NK cells could represent an appealing solution for almost all the above-mentioned hurdles. Indeed, CAR-NK cells do not require HLA matching to be cytotoxic and can be used in allogeneic settings without causing GvHD, thus representing a valid system for the generation of “off-the-shelf” products for clinical use [79,102,223,224]. NK cells express activating receptors, such as NCRs, NKG2D, and DNAM-1, that may be engaged synergistically and independently from CAR, triggering NK killing capability and potentially bypassing loss of targeted antigens as a tumor escape mechanism. Moreover, the ADCC ability of NK cells mediated by CD16 expression is an additional tumor-killing strategy [45,225] that could be used in synergy with the CAR antitumor activity (Figure 3). Nowadays, several different sources of NK cells have been considered for the generation of CAR-NK cells, at both preclinical and clinical levels, including NK cell lines (NK-92 [226], KHYG-1 [227], NKL, NKG, YT, etc.), and NK cells from UCB, PB, and, more recently, induced pluripotent stem cells (iPSCs) [228,229] (Figure 3).

The human NK cell line, NK-92 [226], derived from PB of a patient with aggressive non-Hodgkin’s lymphoma carrying several cytogenetic alterations and the integration of EBV DNA, was chosen from different groups as the NK platform since it can be easily expanded under good manufacturing practice (GMP) standards for clinical applications [230] and provides a homogenous NK cell population. Indeed, the unmodified NK-92 line has been approved by the US FDA for use in clinical trials, after its irradiation and before adoptive transfer, to prevent propagation in patients [231], proving its safety in a phase I/II trial [231,232,233]. To date, CAR-NK-92 cells have been extensively investigated preclinically in several models, including arming the cell line with CARs that recognize HER2 [234], CD19 [233,235], CD20 [236], CD38 [237], CD7 [238], CD3 [239], CD5 [240], GD2 [241], EBNA [242], EGFR and EGFRvIII [243], EpCAM [244], mesothelin [228], and CS1 [245]. The clinical application of CAR-NK-92 cells has been restricted. In particular, data are available of only one first-in-men trial based on CAR.CD33 NK-92 cell infusion in three patients with relapsed and refractory AML [246]. This study showed that at doses up to 5 × 10^9^ irradiated cells per patient, no significant adverse effects were observed, along with marginal and transient patient response. To date, the attempt to optimize CAR-NK-92 approaches is currently under evaluation in several clinical trials that include CAR-modified NK-92 for HER-2 targeting in glioblastoma (NCT03383978), BCMA targeting in multiple myeloma (NCT03940833), and CD19 targeting in CD19^+^ leukemia and lymphoma (NCT02892695).

An important question is whether gene-modified NK cell lines represent better CAR effector cells than primary human donor CAR-NK (CAR-dNK) cells, in terms of reproducibility, viability, effectiveness, risk of side effects, and clinical practicality/applicability. Although the formal comparison of the functional activities of sorted CAR-NK cells generated using the NK-92 cell line with those generated from CAR-dNK cells was recently conducted in an vitro model, demonstrating that CAR-NK-92 cells had stronger cytotoxic in vitro activity against leukemia cells compared to CAR-dNK cells [247], various evidence suggests that NK-92 cells could not be the best NK recipient for CAR-engineering. Indeed, irradiated NK-92 cells have no possibility to expand in vivo after the infusion, the lack of CD16 strongly impairs ADCC, and the lack of NKp44 expression [248,249] compromises the natural cytotoxicity in comparison to activated primary NK cells.

UCB-derived primary NK cells have been explored as a possible platform for CAR approaches for several reasons, including low risk of viral transmission from donor to recipient, rapid availability of UCB units serving as an immediate “off-the-shelf” product, less stringent requirements for HLA matching, and lower risk of GvHD [250]. In vitro models have been developed to optimize the GMP expansion of large-scale UCB-NK cells, using artificial antigen-presenting cells (aAPCs) expressing several costimulatory molecules in association to either membrane-bound IL-21 [180,251] or membrane-bound IL-15 [252]. This last approach has been considered to generate CAR.CD19 NK cells that also produce soluble IL-15 to boost in vivo expansion and persistence, as already demonstrated in a preclinical model [253], and is now under clinical evaluation at M.D. Anderson Cancer Center (recruiting trial NCT03056339 and not yet recruiting trial NCT03579927). Early data on the clinical efficacy of CAR-NK therapy suggest that UCB-derived NK cells transduced with CD19 CARs can be used safely and effectively as off-the-shelf products in patients with B-cell malignancies (oral communication at IACH 2018 meeting. link: http://cme-utilities.com/mailshotcme/IACH/Summaries/Rezvani%20Innate%20Killer%20meeting_2018.pdf). Beside the great advantages in the use of UCB as a source for NK cells, the major limitation is represented by the fact that UCB contains between 10- and 100-fold fewer nucleated cells than other sources of NK cells [250], limiting the amount of cells of interest that can be retrieved from one UCB unit for the generation of off-the-shelf CAR-NK cell banks. This is the reason why we, and other groups, are currently investigating the feasibility and efficacy of CAR-NK cells derived from PB of healthy donors. Several attempts have been conducted in order to obtain large numbers of NK cells from PB sources [132,254,255,256,257,258,259], whereas few of them were associated with the generation of CAR-NK cells. In particular, the feeder expanded approach based on the K562 cell line modified to express membrane-bound IL-15 and 41BB ligand [260] was considered for the manufacturing of PB-derived CAR-NK cells in at least two pilot clinical trials (NCT01974479 in Singapore and NCT00995137 in Memphis USA). Recently, an innovative strategy to generate CAR-NK cells without the use of a feeder layer has been described. In particular, this last approach has been tested in the model of CAR.CD19 NK cells by our group and represents a great advantage in terms of GMP manufacturing as well as safety requirements [261]. The feeder-free, bovine serum-free protocol is based on the ex vivo stimulation of NK cells by monoclonal Ab directed against NCRs to generate high-purity, functional, and expandable PB-NK and PB-CAR-NK cells from widely available donor-derived leukapheresis products or PBMCs. The CAR-NK cells express a broad number of relevant NK cell markers and receptors, indicating that the established method is able to genetically modify and expand heterogeneous NK cells, regardless of their maturation stage and cytokine-induced activation [261].

Stem cells (i.e., CD34^+^ hematopoietic progenitors from peripheral blood and UCB, as well as iPSC) offer another renewable source of CAR-NK cells that can be standardized as an off-the-shelf therapy. While the generation of NK cells from CD34^+^, human embryonic stem cells (hESCs), or iPSC has been largely investigated by different approaches, the ex vivo expansion of CAR-NK cells from these sources has been limited to few reports. In particular, CAR-NK cells have been generated from HSC derived from UCB by using an optimized protocol based on the ex vivo expansion of nonirradiated murine OP9-DL1 stroma in the presence of IL-7 and IL-15 [262]. This approach was feasible, although no scalability proof has been conducted so far to prove its applicability in the generation of CAR-NK off-the-shelf cellular banks. This latter approach, indeed, was carried out by groups working on CAR-NK derived from iPSC [228,229]. The use of iPSC-derived NK cells is currently under early clinical evaluation for safety and feasibility (NCT03841110), paving the way to the iPSC-CAR-NK approach in the near feature.

Of notice, it is also relevant to consider that up to now, most CARs were not optimized for NK cells, saving the CD3ζ domain. One attempt to completely substitute this region, if of any advantage, is represented by the construction of a chimeric molecule between the extracellular region of the inhibitory receptor PD-1 and the transmembrane domain of the activating receptor NKG2D to reverse the immune escape mediated by PD-1 ligands in solid tumors [263]. The authors identified a chimeric PD1-NKG2D receptor containing a NKG2D hinge region and 4-1BB costimulatory domain to obtain stable surface expression and to mediate in vitro cytotoxicity of NK92 cells against various tumor cells [263]. A second approach was to exploit DAP12, a signaling adaptor molecule involved in signal transduction of activating NK cell receptors, fused to the anti-prostate stem cell Ag (PSCA) scFv(AM1) to confer improved cytotoxicity to the NK cell line, YTS, against PSCA-positive tumor cells [264], or fused to the extracellular domain of NKG2D itself [265]. This approach is of particular interest, since it was tested in a pilot clinical trial in three patients with chemotherapy-refractory metastatic colorectal cancer to evaluate the safety and feasibility of adoptive cell therapy with primary feeder-expanded NK cells modified by mRNA electroporation. Patients received multiple infusions of autologous engineered CAR-NK cells (first patient) or allogeneic CAR-NK cells from HLA-haploidentical family donors (second and third patients) with no dose-limiting toxicities or serious adverse effects. Importantly, only grade 1 CRS was reported, associated with fever, fatigue, and anorexia, whereas GvHD was not observed in the two patients treated with haploidentical NK cells [265].

Independently from the applied source for CAR-NK cells and the CAR design itself, the level of CAR expression has been one of the major issues associated with the clinical translation of the approach. Retroviral and lentiviral transductions are the two major platforms used for the stable expression of CAR in NK cells, whereas RNA electroporation approach, providing transient CAR expression, was adopted as a risk mitigation strategy by several groups [266,267]. Moreover, the comparison between lentiviral and mRNA electroporation of CAR in NK cells has been formally conducted, with the great advantage of mRNA electroporation only in the NK-92 cell line, whereas primary NK cells could be efficiently transduced only upon high title lentiviral exposure [268].

Besides all the preclinical achievements in the field of CAR-NK cells, further research, as well as pilot clinical trials (Table 1), are needed to investigate efficacy and feasibility of this novel and intriguing approach, in an attempt to build a novel concept of tailored therapy, in which allogenic effector cells could be used to maximize CAR cell therapy.

## 6. Concluding Remarks

The improved knowledge on the NK cell biology has led to increased interest in the development of different immunotherapeutic approaches based on the use of these cells, and numerous studies have been conducted to exploit their powerful antitumor activity. In particular, the potent cytotoxicity of NK cells has been employed to treat hematological malignancies using two different approaches: (1) adoptive transfer of mature NK cells; and (2) haplo-HSCT, where mature donor NK cells are generated in vivo from HSCs and/or transferred with the graft (Figure 4).

Moreover, the in vitro or in vivo activation of NK cells with immune stimulants (including cytokines, BiKE, and TriKE) or the genetic modification of NK cells with CAR constructs specific for tumor antigens are promising strategies for potentiating and redirecting NK cell response against tumor cells. A combined use of some of these approaches may represent a novel strategy for cancer immunotherapy against hematological malignancies (Figure 4).

New perspectives can be represented by nanosized extracellular vesicles (EVs) that can be naturally secreted by several cell types including NK cells. The NK EVs contain lytic proteins, showing cytotoxic effects on different malignant cell lines, including ALL. They can also transfer bioactive molecules, easily passing through biological barriers. Thus, NK EVs have been recently proposed as a new cell-free immunotherapeutic tool [141,269].

## Figures and Tables

**Figure 1 jcm-08-01702-f001:**
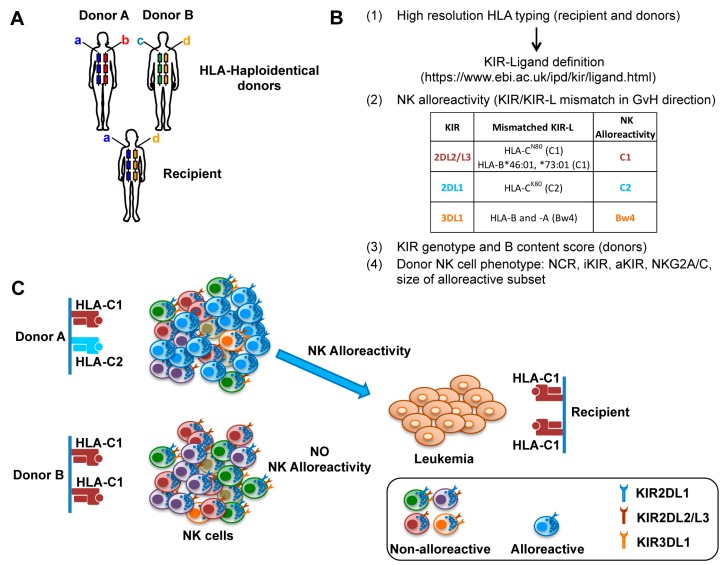
Donor selection in haploidentical-hematopoietic stem cell transplantation (haplo-HSCT). (**A**) Alternative human leucocyte antigen (HLA)-haploidentical donors (e.g., both parents of a pediatric patient) can be available for haplo-HSCT to cure leukemia patients. (**B**) Various analyses can be performed to define the possible donor natural killer (NK) alloreactivity, *killer immunoglobulin-like receptor* (*KIR*) genotype, and NK cell phenotypic repertoire to guide the choice for selecting the optimal donor. (**C**) A schematic representation of donor A and donor B NK cell repertoires, characterized by presence or absence of NK alloreactivity, respectively. Only donor A has alloreactive NK cells, namely “educated” NK cells expressing only KIR2DL1, the inhibitory KIR (iKIR) specific for HLA-C2 epitope, present in the donor and absent in the recipient. The alloreactive NK cell subset of donor A will be highly efficient in leukemia killing, indicating that donor A can be better than donor B.

**Figure 2 jcm-08-01702-f002:**
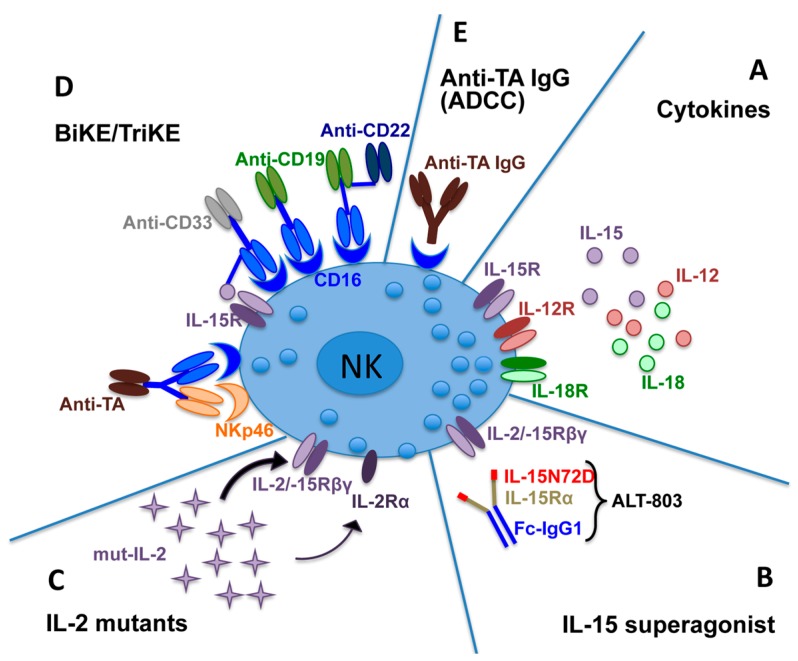
Different strategies can be used to induce activation and expansion of natural killer (NK) cells to improve tumor killing. (**A**) Cytokines, such as interleukin (IL)-15, IL-12, and IL-18 to generate cytokine induced memory-like NK cells (CIML-NK); (**B**) IL-15 superagonist ALT-803, an IL-15 mutant (IL-15N72D) bound to a soluble, dimeric IL-15Rα Fc fusion protein (IL-15Rα-Fc); (**C**) IL-2 mutants with high affinity for IL-2Rβγ present on NK cells, but reduced affinity for IL-2Rα expressed on Treg cells; (**D**) activating NK cell receptor engagement by the use of bispecific or trispecific killer engagers (BiKE or TriKE), capable of binding CD16 on NK cells and one/two tumor antigen(s) (e.g., CD19, CD22, CD33) or CD16 and NKp46 on NK cells and one tumor antigen; (**E**) tumor-specific antibodies (IgG), capable of inducing the NK-antibody dependent cell-mediated cytotoxicity(ADCC). TA, tumor antigen.

**Figure 3 jcm-08-01702-f003:**
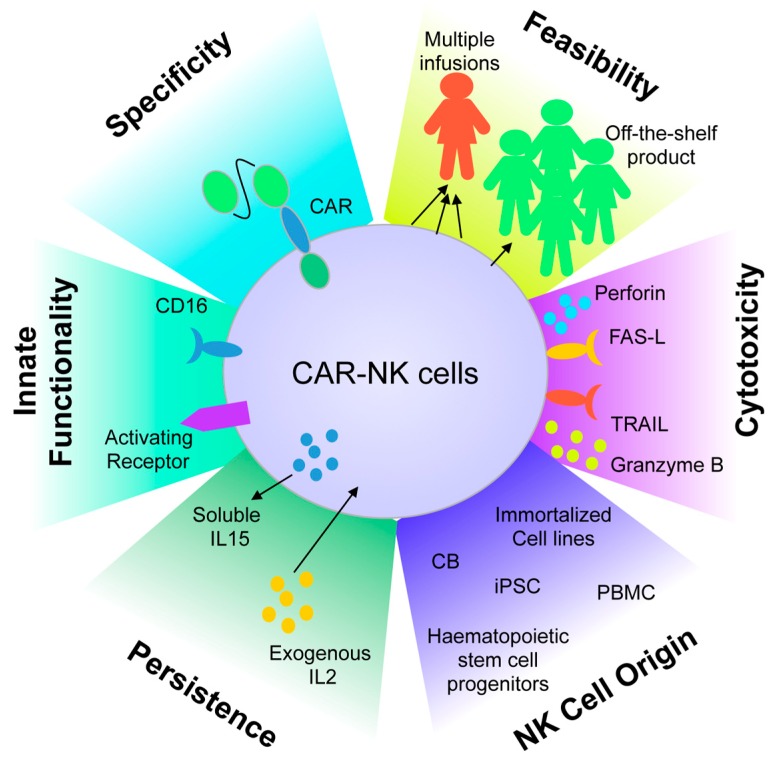
Benefits of chimeric antigen receptor-natural killer (CAR-NK) cells. NK cells of different origin can be genetically modified through the use of CAR constructs able to redirect their specificity against antigens expressed on tumor cells. These NK cells can be further expanded ex vivo to reach clinically meaningful numbers, and further optimized by the activation of their native receptors, including CD16 for the antibody dependent cell-mediated cytotoxicity (ADCC) mechanism. CB, cord blood; iPSC, induced pluripotent stem cells; PBMC, peripheral blood mononuclear cells; FAS-L, FAS-ligand; IL2, interleukin-2; IL15, interleukin-15; TRAIL, tumor necrosis factor (TNF)–related apoptosis-inducing ligand.

**Figure 4 jcm-08-01702-f004:**
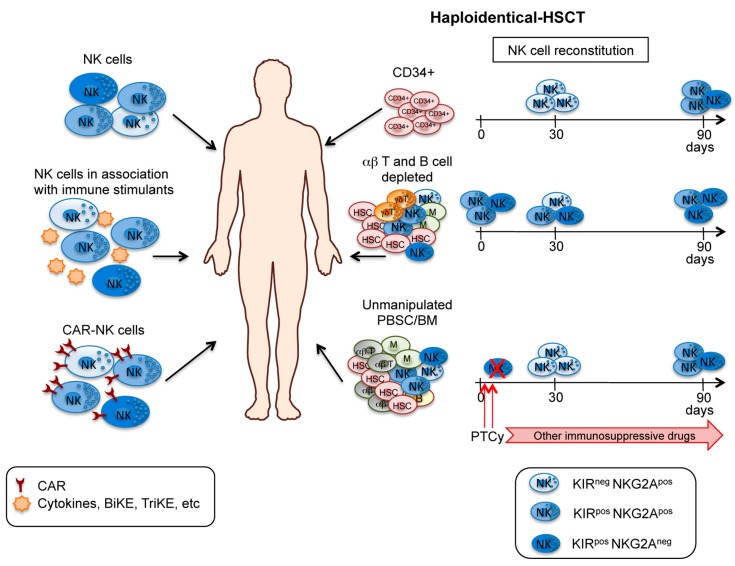
Different therapeutic approaches based on the use of natural killer (NK) cells. Adoptive transfer: infusion of unmodified allogeneic NK cells (directly or in combination with different types of immune stimulants) or chimeric antigen receptor (CAR)-modified allogeneic NK cells. Different strategies of haploidentical-HSCT: graft inoculum of “megadoses” of highly purified CD34^+^ cells; infusion of a αβT- and CD19 B cell-depleted graft enriched for hematopoietic stem cells (HSC) and also containing other cell types, including mature (possibly alloreactive) NK cells and γδT lymphocytes; infusion of unmanipulated peripheral blood stem cells (PBSC)/bone marrow (BM) and early (+3 +5 day) post-transplant high-dose cyclophosphamide (PTCy) administration that eliminates donor-derived proliferating cells, including all mature NK cells. Graft versus host disease (GvHD) prophylaxis is given only in the third type of transplant. NK cell reconstitution in the three haploidentical-hematopoietic stem cell transplantation (haplo-HSCT) platforms is depicted, differentiating different stages of maturation. Only in αβT and CD19 B cell-depleted graft are mature NK cells infused and persist in the recipient. BiKE, bispecific killer engagers; TriKE, trispecific killer engagers.

**Table 1 jcm-08-01702-t001:** Active clinical CAR NK cell trials with a known status (source: ClinicalTrails.gov).

Identifier	NK Type/Source	CAR Target	Conditions	Phase	Status	Last Up-Date	Location
NCT03415100	Autologous or allogeneic NK cells	NKG2D-Ligand	Metastatic Solid Tumours	I	Recruiting	August, 2018	China
NCT03056339 NCT03579927	Primary NK/CB	CD19-IL15	B Lymphoid Malignancies	I/II I/II	Recruiting Not yet recruiting	July, 2019/October, 2019	USA
NCT03692767	ND	CD22	Relapsed Refractory B cell Lymphoma	I	Not yet recruiting	January, 2019	ND
NCT03690310	ND	CD19	Refractory B Cell Lymphoma	I	Not yet recruiting	January, 2019	ND
NCT03692663	ND	PSMA	Castration-Resistant Prostate Cancer	I	Not yet recruiting	October, 2018	ND
NCT03824964	ND	CD19/CD22	Relapsed and Refractory B Cell Lymphoma	I	Not yet recruiting	January, 2019	ND
NCT03692637	ND	Mesothelin	Epithelial Ovarian Cancer	I	Not yet recruiting	January, 2019	ND
NCT03940833	NK-92	BCMA	Relapsed/Refractory MM	I	Recruiting	May, 2019	China
NCT03940820 NCT03931720	ND	ROBO1	Metastatic Solid Tumours/Malignant Tumors	I I/II	Recruiting	May, 2019	China
NCT03941457	ND	ROBO1	Pancreatic Cancer	I/II	Recruiting	May, 2019	China
NCT03383978	NK-92	HER-2	Glioblastoma	I	Recruiting	May, 2019	Germany

ND: Not Defined.

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
