# Peer review of "NK Cell-Based Immunotherapy for Hematological Malignancies"

_jcm, 2019, doi:10.3390/jcm8101702_

Round 1
Reviewer 1 Report
Comments and Suggestions for Authors
Synopsis
The authors review Natural Killer (NK) cell based immunotherapy for hematological malignancies. Different types of hematological tumor-associated NK cells and their anti-tumor capacity and ability to modulate other anti-tumor responses are described. The current understanding of NK cell biology and the interest in NK-cell -based antitumor therapies and usage in clinical practice are reviewed. Immune-stimulatory molecules can either selectively enhance NK cell activity while maintaining their in vivo survival and proliferation (cytokines), or mediate NK cell cytotoxicity by ADCC ( antibodies). ex vivo expansion/activation technology, as well as new approaches to genetically modify NK cells, adoptive transfer of NK cells holds promises to become powerful in the fight against cancers. The authors surmise that an in depth understanding of biological and functional changes in tumor-associated NK cells and their impact on overall tumor control is a very important given the increasing interested in NK cell-based therapeutic approaches to cancer treatment.
Comments to authors
This is an interesting and highly topical review that exploits the NK cells high potential in clinical practice and their anti-tumor responses in hematological malignancies. As the authors highlight in their manuscript, the rising interest in NK cells and their application in cell-based cancer treatments necessitates a fundamental understanding of how tissues/tumors impact NK cells and their ability to kill tumor cells or regulate immune responses. A solid synthesis of the most current literature is provided. However, the review would benefit from a brief expansion on NK cell plasticity as well as the difference between NK cells and ILC1 in general consideration that is not mentioned in all the manuscript. With the above in mind, the following comments should be addressed in order to give a clearer and more comprehensive overview of the current understanding of how biological and functional changes in hematological tumor-associated NK cells may impact immunotherapy.
NK cells counts are associated with Molecular Relapse-Free survival after Imatinib discontinuation in Chronic Myeloid Leukemia (CML). The authors should mention the IMMONOSTIM study and the role of NK cells in CML together with AML. The authors should add a table in the manuscript in the section of CAR, about the current clinical trials of CAR-NK cells. The cost for producing, preserving and transporting clinical grade living immune cells is high, posing a challenge to wide applications in clinics. In recent years, EVs, a nano-sized vesicles naturally secreted by many different types of cells including NK cells, have gradually been proposed and studied, providing a new cell-free immunotherapy avenue. Can the authors discuss this newly and innovative approach? The authors should discuss these two open questions: i) what is the clinical relevance of CMV‐induced NKG2C+memory NK cells in treating leukemia?And ii) Is therapeutic efficacy of memory NK cells limited to single type of AML or is it extendable to B‐and T cell ALL? Will it be equally effective in adult and pediatric leukemias?
Reviewer 2 Report
This is a well written and extensive review that may be interesting to many investigators in the field of cancer immunotherapy. I have only few minor comments.
Authors describe in detail several NK cell inhibitory and activating receptor systems. What I think is missing it is lack of information about an important NKRP1A inhibitory receptor and its LLT1 ligand. There is a growing bulk of evidence that this receptor may play a crucial role in tumour escape from under NK cell control. Thus it would be of interest to put some information regarding this putative NK cell checkpoint. (line 90) It will be of interest to mention that one of the "long" KIRs, KIR2DL4 is not truly an inhibitory one. Although it posses one ITIM segment it also has a positively charged aa that recruits adaptors with ITAM thus making KIR2DL4 a stimulatory receptor. (line 181) A correct name for TGF is Transforming Growth Factor. (line 420) In line with my previous query on NKRP1A it may be worth of adding that stimulatory effect of IL2 on NK cells may also involve downregulation of this inhibitory receptor. Some paragraphs are too long that makes the content hardly digestible. The authors use "pos" or "neg" to indicate positive or negative expression, respectively. It is not a mistake but it is rather commonly accepted to use "+" or "-".
